# Eyeing ID: Access to Identification as a Barrier to Banking and Other Social Determinants of Health

**DOI:** 10.3390/ijerph22101552

**Published:** 2025-10-12

**Authors:** Katie Bonner, Natalia Fana, Sarah Lunney, Sarah Campbell, Deanna Merriam, Cristian Estrella Almonte, Sarah Gander

**Affiliations:** 1Horizon Health Network, Fredericton, NB E3B 3B7, Canada; kab235@mun.ca (K.B.); natalia.fana@horizonnb.ca (N.F.); sarah.lunney@horizonnb.ca (S.L.); deanna.merriam@horizonnb.ca (D.M.);; 2Saint John Campus, University of New Brunswick, Fredericton, NB E3B 5A3, Canada; 3Faculty of Medicine, Saint John, NB Campus, Dalhousie University, Halifax, NS B3H 2Y9, Canada

**Keywords:** personal identification (ID), lack of ID, vulnerable residents, ID policies

## Abstract

Personal identification (ID) is a prerequisite to many financial and social services; however, many vulnerable residents do not have ID and lack the resources to acquire it. To assess the impact of ID inaccessibility in a local context, a study was conducted throughout New Brunswick, Canada. The study objective was to understand the implications of ID requirements and the barriers to acquiring it through the lens of consumers. This mixed-methods, observational study included surveys and interviews. The survey collected demographics, socioeconomic status (SES), financial behaviors and experiences, and barriers to accessing ID. The semi-structured interviews explored individual experiences. In order to address disparities in health and social outcomes, ID requirements and barriers to access need to be acknowledged and mitigated. A total of 142 surveys were completed. Many respondents reported difficulty obtaining or replacing a driver’s license (30.8%), a provincial photo ID (47.7%), or their birth certificate (39.4%), identifying cost (34.4%) and required documentation (28.1%) as the main barriers. Thematic analysis identified three main themes: the difficulty of living without ID, barriers to obtaining or replacing an ID, and an exploration of solutions. Current ID policies restrict access to community services such as banking, housing, and employment, which are intended to support individuals to improve their situation and gain autonomy. Policies and services are required to address this urgent issue.

## 1. Introduction

Personal identification (ID) such as a birth certificate, driver’s license, passport, Medicare card, social insurance number (SIN), or other government-issued photo ID is a common requirement for accessing many public and private services. While ID can facilitate privacy, security, or eligibility screening, it can also create significant barriers for individuals without this necessary documentation, particularly those experiencing poverty, housing instability, or other forms of marginalization [1,2]. Obtaining or replacing ID typically requires access to existing documents, literacy to navigate bureaucratic systems, and financial resources to pay the associated fees [3,4]. For individuals who are precariously housed, there may be no secure place to store ID, and replacing lost documents can be unaffordable or require forms of ID they no longer possess. These challenges form a cycle of exclusion in which ID is required to get ID, resulting in inaccessible social services and broader implications of social marginalization [1].

In Canada, ID is frequently required to access essential services such as housing supports, financial services, food banks, social assistance, and healthcare. These services include key components of the social determinants of health (SDOH), and barriers to accessing them directly impact health and well-being [2,5]. Individuals without ID face restricted access to income supports and difficulty securing housing and can be excluded from food distribution programs that require proof of residency or government-issued documents [6,7]. Restrictive ID policies can also result in a disproportionate negative impact for people who are undocumented, formerly incarcerated, racialized, or living with unstable housing [2,8]. LeBrón et al. [2] argue that ID policies must consider equity embedded in their design and implementation to ensure marginalized groups are not continuously excluded from basic services that can contribute to ongoing toxic stress, trauma, and therefore worsening chronic conditions. In this way, a lack of ID functions as a form of social exclusion, reinforcing poverty and compromising health outcomes for some of the most vulnerable members of society. As such, ID inaccessibility is an environmental and structural determinant of health equity.

A 2020 review by Sanders et al. [1] found that the acquisition and retention of ID are primary barriers to accessing health and social services in North America. Their review highlighted how restrictive ID policies disproportionately affect Indigenous communities, people experiencing homelessness, and individuals with complex social needs. In a related case study, Sanders and Burnett [5] described the intergenerational impacts of unregistered births among Indigenous populations in Northern Ontario, where entire families may lack essential ID documents like birth certificates. In response to these ID access concerns, some communities have developed ID programs to assist individuals in completing paperwork, store documents safely, or offer alternative biometric verification methods such as fingerprinting [3,9,10,11]. Vancouver’s Kettle Society, for example, offers an “ID Bank” where low-income, homeless, and/or marginally housed individuals can safely store important documents (such as birth certificates, citizenship certificates, permanent residency cards, passports, and more) for free [12].

In New Brunswick, Canada, the process of obtaining ID includes several logistical and financial barriers. To get a provincial photo ID (i.e., driver’s license), individuals must appear in person at a Service New Brunswick (SNB) Centre during business hours and present multiple pieces of documentation, including proof of name, date of birth, and residency [13]. Many of these required documents (such as a birth certificate) must be acquired beforehand at additional cost, creating a compounding barrier to accessing ID. For example, as of 2025, a New Brunswick photo ID card costs CAD48.00, and replacing it costs USD 15.00. A birth certificate costs an additional payment of CAD 45.00 [14]. These expenses may create a financial barrier for those receiving income assistance or those without stable employment, who often require ID to access additional services to help them address the SDOH.

Access to many essential services hinges on accessing ID documents. In Saint John, New Brunswick, Canada, food banks require clients to present a valid Medicare card and proof of residency, including “a utility bill, bank statement, insurance forms, rent receipt, or an I.D. with your current address” [15]. Without a lease or utility bill, which are often unavailable to people without housing, this requirement can be impossible to fulfill. Similarly, local housing services such as the Coordinated Access List (CAL) require SIN numbers and proof of income, reinforcing the exclusion of individuals who lack ID or those living with low SES.

Additionally, researchers have increasingly recognized financial health—an individual’s capacity to control spending, anticipate and recover from financial shocks, have little debt, and plan for future goals—as an underappreciated social determinant [16]. Brown et al. [17] and the OECD [18] emphasize the strong correlation between financial well-being and self-rated health, stress levels, and access to basic needs. Individuals without ID may be unable to open bank accounts or deposit government cheques, creating dependency on informal or predatory financial services such as payday lenders [19]. Financial inclusion is defined as having sustained access to financial products and services that are required by the individual. [20] Income is directly related to the ability to access many other important social determinants of health, including food and transportation to access essential services. Programs that address a raise minimum wage and social assistance approach the provision of a living wage that promotions inclusion and a higher potential to thrive [21]. A living wage is defined as an hourly rate needed to provide and live with dignity. It is calculated based on a family with two adults and two children, based on local costs and reevaluated annually. In the New Brunswick October 2024 report calculated by the Human Development Council, the living wage was CAD 24.62/hour while the minimum wage was CAD 15.30/hour [22]. As mentioned, the ability to be securely housed rests largely on income and the availability of affordable places to live. Certain policy changes have left too many New Brunswickers living in shelters, couch surfing, or on the streets. As such, it is a powerful determinant of health, further isolating and compromising the ability to take action to improve health and quality of life [23]. Many cities in Canada have large urban sprawls with poor planning for public transportation. Canada also has a large rural population. This presents an additional challenge and need for individuals to have a reliable income and be able to access transportation as a determinant of health in terms of being able to improve access to health care and community resources [24].

These structural challenges illustrate the urgent need to understand the real-life barriers people face when trying to obtain or replace ID. This study seeks to explore these barriers from the perspective of those most impacted, evaluate how ID access intersects with financial exclusion, and assess participant perspectives of solutions that could reduce harm and improve inclusion, such as ID Banks or Biometric IDs.

## 2. Materials and Methods

The current study is an observational study of the barriers created by IDs, barriers to ID acquisition, and perceptions of proposed solutions to these barriers. An Advisory Panel of local stakeholders and industry professionals informed our methodology and provided insight regarding relevant indicators. The study utilized both quantitative and qualitative data derived from surveys and semi-structured interviews. Participants in the study were individuals who self-identified as experiencing barriers to accessing ID and financial inclusion. All study participants voluntarily consented and were required to be 19 years of age or older, living in the province of New Brunswick, as per the regional Horizon Health Network Research Ethics Board (REB) approval.

### 2.1. Recruitment

Participant recruitment took place through partnerships with local community organizations that work with community members experiencing barriers imposed by a lack of ID and/or barriers to obtaining and/or retaining ID. Partner organizations that work with community members in poverty allowed the use of selective recruitment methods to target individuals who could be potentially impacted by the topics under study. As a result, the consumer experiences discussed in this study are primarily from urban communities in New Brunswick, Canada.

The survey was advertised online at (www.nbsocialpediatrics.com and www.loanfund.ca, accessed on 1 February 2021) and through social media channels (Facebook and Twitter). It was also advertised through Around the Block (a community newsletter) and other media outlets. CAD 10 gift cards were provided as an incentive to participate (limit 2 per household) in the survey and/or interview. Those wishing to receive the gift card needed to provide a mailing address to receive the card if completing the survey online. These gift cards were purchased by the research team and mailed to the participant. For in-person participants, gift cards were available on site.

### 2.2. Surveys

The surveys were available online or in person with a member of the research team who would then enter survey responses. In-person surveys were entered into the Survey Monkey platform by a member of the research team who assisted the study participant with their in-person survey. Surveys were administered through SurveyMonkey. Consumers who consented to the study completed a survey that collected information on demographics, socioeconomic status, financial behaviors and experiences, and barriers to accessing ID. Survey questions regarding financial behaviors and experiences were adapted from the Organization for Economic Co-operation and Development/International Network on Financial Education (OECD/INFE) Financial Literacy survey [18], which was also used to evaluate financial attitudes. Surveys included a combination of multiple-choice, true/false, and open-ended questions.

### 2.3. Interviews

A small number of consumers who completed the survey also participated in a semi-structured interview. Interviews were conducted either in person (following all COVID-19 safety protocols in place at the time) or over the phone. All interviews were recorded and transcribed verbatim. The interviews contained open-ended questions designed to discuss the experiences of participants obtaining ID, accessing services without ID, their attitudes regarding biometric ID and ID banks, and potential mechanisms to make ID and services more accessible.

### 2.4. Data Analysis

Quantitative survey response data was extracted from the Survey Monkey platform as an Excel file. The data was checked by a member of the research team for any missing values or inconsistencies. Quantitative survey response data was uploaded as an Excel file to SPSS version 22 for analysis. This was primarily a descriptive analysis of survey results. The questions used to assess the financial attitudes of respondents were based on the OECD’s 2018 survey and were scored according to the survey’s protocol [18]. Participant Forward Sortation Area (FSA) (the first three digits of Postal Code) data was also collected from survey respondents to conduct a geographic analysis of the neighborhood in which they reside. FSA data was analyzed through Statistical Analysis System (SAS) 9.4 by using the Postal Code Conversion File Plus (PCCF+) to access relevant geographical and Census data associated with the FSA. Socioeconomic status was scored on a scale of 0 to 8 to determine the SES of survey respondents.

Qualitative thematic analysis was conducted on the interview transcription data and open-ended survey questions. Initial coding was manually conducted by several members of the research team independently following Braun et al.’s 2018 thematic analysis framework [25]. This included the identification of key ideas and phrases in the text. The research team then met to discuss the first-level codes and group them into themes through consensus. Once these themes were named and defined, a thorough analysis was written for each, with participant quotations used as supporting evidence.

## 3. Results

### 3.1. Consumer Experience: Survey Results

In total, 142 surveys were administered to participants in New Brunswick. The survey was administered primarily digitally, with paper copies available to community members who may not have otherwise had the means to complete the online survey. When paper copies were completed, a member of the research team input survey responses into the digital survey as they appeared on paper.

#### 3.1.1. Demographics and SES

Scoring socioeconomic status resulted in an average SES of 1.13 (*n* = 119), with scores ranging from 0 to 4 on a scale of 0 to 8. Scores between 0 and 3 represent low SES, 4 to 6 represent moderate SES, and 7 to 8 represent high SES. As recruitment efforts targeted low-income individuals, these low SES score results represent a restricted range of participants rather than the general population.

#### 3.1.2. Financial Attitudes and Behaviors

Using the OECD Toolkit for Financial Literacy to guide the development of survey questions, participants were asked about their financial behaviors and experiences. Over two-thirds of respondents (69.8%) agreed that their financial situation limits their ability to do things that are important to them, and 82.6% worry about paying their normal living expenses sometimes, often, or always. Overall, only 18% said they were satisfied with their current financial situation, with 51.5% reporting that they never or rarely have money left over at the end of the month. Overall, 74.6% at least somewhat feel like they will never have the things they want in life because of their money situation.

Financial attitude scores were calculated on a scale of one to five based on the scoring guide available in the OECD/INFE Financial Literacy Toolkit [18]. The financial attitudes scoring asks participants to indicate their level of agreement with statements relating to long-term financial planning and saving. The minimum target score is >3, and the global average is 3.0 out of 5. Of the survey respondents, the average financial attitudes score was 2.80, while only 49.6% (*n* = 70) of respondents scored 3 or greater.

Three-quarters of participants (75.4%) indicated that in the last 12 months, they found their income did not quite cover their living expenses. When asked how they made ends meet last time this happened, 86.1% of respondents said they used community resources (such as food banks or soup kitchens), 78.4% did without, paid bills late, or missed payments, 73.8% borrowed money from family or friends, and 65.4% sold or pawned their belongings. Overall, 27.5% of respondents used overdraft, line of credit, or a credit card to make ends meet, and 9.7% took out a payday loan.

#### 3.1.3. Banking Experiences

Participants were asked about their current banking habits and experiences. Over 50% of participants (53.3%) currently hold a savings account with a bank, and 76.8% possess a checking account. One quarter (25.5%) currently owns a credit card. When participants who did not have a chequing or savings account were asked about the reason why, the main influencing factors were reported to be a lack of identification (38%), bank fees (21%), no need (18%), and not eligible (17%).

#### 3.1.4. Experiences with ID

When asked about which types of ID they had on hand, or readily available in a known location, 68.2% (*n* = 88) of participants had a birth certificate, 61.7% (*n* = 82) had a driver’s license or other government issued photo ID, 20.9% (*n* = 27) had a passport, while 16.7% (*n* = 12) had no ID at all. Other forms of ID mentioned by respondents included Medicare cards, social insurance numbers, and status cards. Many respondents (73.7%, *n* = 98) indicated that they had lost their ID at some point in their lives. When participants were asked about their experiences obtaining or replacing a piece of ID, 30.8% (*n* = 40) said they had difficulty obtaining or replacing a driver’s license, 47.7% (*n* = 61) had difficulty replacing a provincial photo ID, 39.4% (*n* = 52) had difficulty obtaining or replacing their birth certificate, and 13.4% (*n* = 17) had difficulty with a passport.

Of the 90 participants who identified difficulty with the process of replacing an ID, the main barriers identified were cost (34.4%, *n* = 33) and required documentation (28.1%, *n* = 27).

#### 3.1.5. Perceptions of Biometric or ID Bank Alternatives

When participants were asked what they would prefer to show as a form of ID, the most popular option was government-issued photo ID, which 50.0% (*n* = 60) of respondents preferred. The second most popular response was fingerprint or eye scan (33.3%, *n* = 40).

Three open-ended questions were asked at the end of the survey, giving participants the opportunity to expand on topics discussed throughout the survey. When asked what improvements they would like to see in terms of getting an ID, 27.6% of participants mentioned cost, some stressing that if ID is a requirement for so many services, particularly government services, they should be free to obtain. Rather than saying it should be free altogether, one participant addressed the cost concern with the suggestion that it may be more accessible if consumers did not have to pay the full cost up front but rather could make payments, pay once they receive their social assistance cheque, or receive a discounted price. Other participants noted that social assistance could perhaps help cover associated costs. Common themes included making ID easier to acquire, shorter wait times, temporary ID cards or ID alternatives while waiting to receive documentation in order to get an official ID, a free or low-cost ID alternative to government-issued photo ID, and more flexibility with the required proof of identification to apply for ID.

The intergenerational impact of ID barriers was also discussed through these open-ended survey questions. One participant described their struggle to obtain an ID for their teenage child because they did not have an official custody agreement with the child’s father, who they had not been in contact with since shortly after their child’s birth: “Getting ID for children without an official custody agreement and/or requiring both parents’ signatures would be awesome. My [child] does not have a passport for this reason as [their] father has not had contact since shortly after [their] birth, but is listed on [their] birth certificate. This has created a barrier as I don’t have money to hire a lawyer to help me figure out what to do to have legal documents listing me as the sole legal guardian.” The generational impact of ID barriers was addressed by participants in both consumer and community partner interviews as an issue that further exacerbates the complexity of the ID acquisition process.

When asked about improvements in terms of accessing services that require ID, a notable theme was the desire for more kindness and understanding. For example, participants desire “more consistency with information from customer service workers (and also more kindness)” and “for people to just be understanding and not just say NO, can’t help you.” Other participants mentioned that they would like to have a better understanding of the ID requirements for various services and how to fulfill these requirements.

### 3.2. Consumer Experience: Interview Results

Seven interviews were conducted with New Brunswickers between the ages of 30 and 65. Participants’ main sources of income were identified as social assistance, employment insurance, the Canada Emergency Response Benefit (CERB), part-time employment, and family. Some participants did not have an ID at the time of the interview, and others did not have bank accounts. A thematic analysis of consumer interviews provided further insight into ID as a barrier to social service delivery and perceptions of potential solutions. The three main themes identified were (1) living without an ID, (2) obtaining or replacing an ID, and (3) solutions to ID barriers (Figure 1).

#### 3.2.1. Living Without ID

Interview participants reported experiences of losing ID in a number of different ways, such as through a house fire, theft, or simply having their ID fall out of their pocket while riding a bicycle. Service denial resulting from lack of ID was primarily related to bank access, with one participant noting, “yeah ‘cause you lose everything and got no ID. They don’t want to give you a bank card if you got no ID. Even if you got a bank out there, you’re stuck… your money’s there, you’re stuck to not get it out of there.” Another participant indicated that the bank would not even let them cash their government assistance cheque without an ID. Other impacts of not having ID included an inability to travel and inability to participate in social activities: “I can’t go over the border if we didn’t have COVID. if I had my IDs, I would be able to go see family members. My dad is pretty much on his death bed and I cannot go visit because I don’t have ID”, “Um… as well, if I was to go to… I don’t… but if I was to go to a bar or to the liquor store or anything that… where you need ID, I would not be… even though I’m 35, I still would not be able to go cause I don’t have ID.”

To cope with the issue of not having an ID, participants mentioned that they either go without services that require ID or memorize their numbers (such as SIN or Medicare) while not having a physical card. To access banking services like cheque cashing, some participants resorted to payday lenders, such as Money Mart. Participants mentioned that when they set up an account with Money Mart initially, Money Mart photocopied their identification documents and kept them on file. By allowing their identification documents to be kept on file to confirm their identity, participants could avail financial services at Money Mart, despite not currently having a physical copy of their ID, by simply providing a phone number associated with their file.

#### 3.2.2. Obtaining or Replacing an ID

When attempting to receive or replace a piece of ID, identified barriers included cost, time, proof of address, and the cyclic complexity of the system. One participant who lost all identification in an accident stated the following: “everything happened in 2013. We’re now in 2021 and that’s why I don’t have any identification… because finances and time.” This participant was born outside of Canada and also noted cost as a significant barrier: “It’s very costly. For my birth certificate… it’s CAD 25. For a… to get a New Brunswick ID, I need my… er my birth certificate as well as my landed immigrant status paperwork. Now that is also very costly. It is CAD 85. And then to get um… to get my Canadian ID, it’s also another CAD 85, so being on Assistance and only having my Assistance and Child Tax, I don’t financially… unable to get any of my IDs back.”

Proof of address was also identified as a barrier. Once a person brings all appropriate documentation to Service New Brunswick to obtain an ID, the ID needs to be mailed to a home address. There is no option to pick it up. For those who are precariously housed, this was identified as a major barrier, as they had nowhere to send their documents safely. As one participant noted, in order to get their ID, some individuals resort to lying about their address, “Like people that are homeless, for example, who want to get like a photo ID you can’t unless you provide an address. And if you can’t provide that, then… and if you do provide one that’s wrong, you’re considered fraudulent so it’s almost as if like…. I don’t know… governments or whatever and keeping the people that way in a sense.” Similarly, it is often required to provide multiple pieces of mail to confirm your address. For those who were precariously housed, this proved next to impossible as they did not have a consistent address where they received mail.

The process of acquiring a piece of ID, whether for the first time or as a replacement, was also discussed as extremely time-consuming. Several participants cited the length of time they waited for certain pieces of ID, ranging from days to months, as a barrier: “I know I got my long form birth certificate and that took six months. Can you imagine?,” “I had to show… I had to show mail and addresses to get it back. I had to go for my picture ID. They just looked on the computer CAD48 later. Then I had to get my Medicare. I had to get a bank card then I couldn’t get my bank card without my picture ID. Uh, yeah. It was a full couple days trying to get it.”, “About a month all together.” When asked about wait times for getting a non-Canadian birth certificate, a participant replied that it took 60 days, and another replied “at least six months” to get their landed immigrant status. When a participant who was adopted from out of province required their birth certificate, it took nearly a year: “With my birth certificate…. Wow, that was a little difficult simply because it shouldn’t have took as long as it did, but due to being adopted from [another province]… they had to find like my birth mother’s information and all that so it took a lot longer… like six months, almost to a year. Like, I forget where I was… maybe eight months, six to eight months so… But I received it, thank God (laughs).”

While proof of address, time, and cost were all identified as barriers, it was clear that the compounding effects of these various challenges were a barrier in themselves. It was noted by one participant that the problem seems to be getting worse, as costs have risen over the past years, and the process has grown more and more complex: “But I mean like, I mean, I’ve noticed things change. Before you could get it like the same day. You have to wait 14 days now and it used to be ten bucks and jumped to forty but… I mean I’m not complaining cause I was helped but I’m sure for the average person out there it could be, you know, overwhelming… you know, like waiting two weeks… like you say if you are homeless and you can’t have it mailed somewhere because they mail it to you I guess. You can’t pick it up so…”. With many participants experiencing the compounding pressures of proving their address, paying for their IDs, and holding on throughout unexpected wait times, these barriers intersected with participants’ various lived experiences to make accessing an ID a complex challenge.

#### 3.2.3. Solutions to ID Barriers

Two participants noted local community resources that helped them obtain their ID: ReConnect, in Moncton, New Brunswick, and the Learning Exchange, in Saint John, New Brunswick, which offer education services to help individuals navigate barriers to employment, healthcare, or social integration [26,27]. Beyond these two groups, there was no mention of other community support that helped participants obtain ID. Two participants mentioned using Money Mart as an alternative to banks in order to cash cheques, simply because Money Mart does not require clients to regularly bring in their ID once they have created a file with them.

When asked about their thoughts on biometric alternatives or ID banks as proposed solutions, feedback was generally positive, with participants appreciating that biometrics would be easier because there is nothing to lose and one would always have ID with themself: “That’d be a good idea because you’d always have ID that way … (laughs) … always got your thumb.”. While most supported the idea, there was also some hesitancy: “Personally, I wouldn’t use it but… So, like that kind of freaks me out… you know… but no, I don’t know. Personally, I wouldn’t like it.” The interviewer then followed up and asked why the participant felt this way: “Um… I would say the privacy thing. You know… I think that’s kind of privacy and there’s just something to me… that’s weird about that… Maybe I’m just behind the times… But no, I find it just weird”.

When asked for their thoughts on ID Banks as a solution, the feedback was generally positive once again, with one participant noting, “I think some people in my neighborhood would feel more comfortable with that” as opposed to biometric solutions. Participants thought the concept of an ID bank was a good idea because it would reduce the risk of losing it, and they could just go obtain their ID during the hours the bank was open. One participant also pointed out that “if you’re driving, they accept a photocopy of your driver’s license too.” Similar to those of biometric solutions, there were some reservations with the idea of ID banks: “Like, I’m so organized… like I like everything with me. You know, so… probably not for me, no”, adding that “it all comes down to privacy. You know… I trust myself before I trust anyone else (laughs) so, yeah.”

## 4. Discussion

This New Brunswick-based study contributes to a growing body of the literature asserting that a lack of personal identification is a critical barrier to accessing essential services and achieving health equity. Our findings demonstrated how cost, documentation requirements, proof of address, and long wait times interact to produce compounding access barriers for individuals, especially for those who are precariously housed or those who demonstrate a low SES. These barriers are not incidental; they exist as structural mechanisms that disproportionately impact vulnerable populations and contribute to the reproduction of social exclusion. These findings align with Sanders et al. [1] and LeBrón et al. [2], who have argued that restrictive ID policies deepen social exclusion by limiting access to healthcare, housing, food, financial services, and other key pillars of the social determinants of health. Our study builds on this scholarship by offering a localized, participant-informed understanding of these barriers in urban New Brunswick settings.

### 4.1. Document and Administrative Concerns

Survey and interview data revealed that over 70% of participants had lost their ID at some point and that cost (34%) and access to required documentation (28%) were the most frequently cited barriers to replacement. Many participants described being caught in a cyclical trap: ID was required to obtain ID; however, replacing lost or stolen documents required other forms of documentation that they did not have, with some noting, “I had to show… I had to show mail and addresses to get it back. I had to go for my picture ID. They just looked on the computer CAD 48 later. Then I had to get my Medicare. I had to get a bank card then I couldn’t get my bank card without my picture ID. Uh, yeah. It was a full couple days trying to get it.” or “people that are homeless, for example, who want to get like a photo ID you can’t unless you provide an address. And if you can’t provide that, then… and if you do provide one that’s wrong, you’re considered fraudulent so it’s almost as if like…. I don’t know… governments or whatever and keeping the people that way in a sense.”. This cycle was intensified by limited service hours, the need for in-person visits, and not having a secure place to store physical documents.

These findings connect with prior studies that emphasize the compounding nature of bureaucratic and financial barriers to ID [3,4,7]. Wusinich et al. [28] describe similar dynamics among unhoused populations, where obtaining legal identification is made more challenging by documentation requirements and inaccessible service processes. Where some of our participants experienced stigma when trying to access services without proper documentation, Dickson-Gomez et al. [29] support this finding by noting how service provider perception contributes to “unofficial” gatekeeping that can deny individuals housing and services even when they technically meet eligibility requirements. These studies point to a broader pattern: administrative structures often function less as neutral service providers and can perpetuate systems of control, determining who is socially included or socially excluded.

Our findings also emphasize that ID inaccessibility can have intergenerational consequences. For instance, one participant could not obtain an ID for their child due to custody complications and the requirement for joint parental consent. This challenge effectively barred their child from accessing school and health supports, showing that a lack of ID can result in intergenerational barriers for those attempting to access the social determinants of health. This aligns with the findings of Sanders and Burnett [5], who document how a lack of documentation (specifically birth certificates) in Indigenous communities has led to the absence of entire family units from social systems and services.

### 4.2. Financial Exclusion

Financial exclusion also emerged as both a symptom of and contributor to ID inaccessibility. Nearly 75% of participants reported that their income did not cover their basic living needs. Many could not open a bank account, cash a government cheque, or create a financial safety net because of ID restrictions. One participant described this struggle, “I need my… er my birth certificate as well as my landed immigrant status paperwork. Now that is also very costly. It is CAD 85. And then to get um… to get my Canadian ID, it’s also another CAD 85, so being on Assistance and only having my Assistance and Child Tax, I don’t financially…. unable to get any of my IDs back” with another participant noting, “yeah ‘cause you lose everything and got no ID. They don’t want to give you a bank card if you got no ID. Even if you got a bank out there, you’re stuck… your money’s there, you’re stuck to not get it out of there”. As a result, several relied on predatory financial services like Money Mart, which do not require the same level of documentation but often impose steep fees and strict lending conditions. While creative, these workarounds expose people to financial exploitation and highlight the urgent need for systemic intervention. Compared to global benchmarks [19], the financial attitudes and behaviors reported in our study reflect an underbanked population navigating structural constraints rather than personal financial illiteracy.

These findings also align with McCabe [30], which examines how poverty governance is enacted through service provider discretion in housing voucher programs. This work noted that decisions about who is able to participate in programs often reflect moral judgments rather than objective need. While participants in our study reported limited financial literacy, their efforts to engage with the financial system demonstrated that systemic barriers were a key obstacle. The OECD [18] and Brown et al. [17] emphasize this link between financial exclusion and health disparities, underscoring the need to treat financial health as a critical component of SDOH.

### 4.3. Perceptions of ID Banks and Biometric Solutions

A key contribution of this study is its exploration of community attitudes toward proposed solutions such as biometric ID systems and ID banks. Overall, participants largely supported the idea of ID banks as a way to reduce the risk of loss and ease access to services, with one participant explaining that between Biometric ID and ID Banks, “some people in my neighbourhood would feel more comfortable” with ID Banks. Biometric options were met with cautious interest: while some participants welcomed the idea of using a fingerprint or eye-scan to verify identity (“That’d be a good idea because you’d always have ID that way … (laughs) … always got your thumb“), others raised concerns about privacy (“it all comes down to privacy. You know… I trust myself before I trust anyone else (laughs) so, yeah.”). These responses align with Smith and Miller [28], who argue that biometric systems must be implemented carefully to avoid undermining civil liberties. Parsell et al. [31] similarly warn that newer state interventions can frame low-income individuals as needing surveillance and behavioral correction, actions that may challenge autonomy and reinforce stigma. These notes align with our observed participant caution and show that technical solutions like ID Banks and Biometric ID may not resolve systemic social exclusion, as solutions must be voluntary, flexible, and rooted in the needs and rights of affected communities.

Currently, however, existing Canadian models that utilize ID banks or biometric ID support this approach. The Kettle Society’s ID bank in Vancouver [11,12] and the ATB Financial biometric ID pilot in Alberta [10] show that alternative ID strategies can be both feasible and well-received when they are designed with trust and accessibility in mind. LeBrón et al. [32] describe a similar initiative in Michigan, where the Washtenaw County ID project provided undocumented and low-income residents with a locally recognized ID to help restore access to services and social participation without state surveillance. Examples also exist outside of Canada, including reports on improved trade practices in Africa [20]. This study concluded that “equity must be embedded in policy design and implementation, which is a long-term commitment. The persistence of inequities in ID acceptance point to the importance of equity being institutionalized, not supported by an outside initiative.” [33] (p. 59), emphasizing the vital role of equitable policy creation that considers the nuanced needs of a community.

### 4.4. Future Implications

The findings of this study point to a need for cross-sector, equity-oriented ID policies that reduce the cumulative burden on marginalized individuals. Addressing cost alone is not sufficient; policymakers must also consider restrictive documentation requirements, community collaboration, and service provider discretion. Government investment in human-centered solutions, such as in ID banks or programs that subsidize or eliminate ID replacement fees, could reduce these barriers and improve access to housing, food, and financial support.

Browne et al. [34] argue that even rights-based homelessness prevention frameworks can be undermined by the very bureaucracies tasked with implementing them. Similarly, our findings suggest that ID frameworks must be redesigned not just to function efficiently, but to respond flexibly to the lived realities of poverty, mobility, stigma, and discrimination as described by our participants. Future research could include feasibility studies for community–government partnerships and pilot initiatives in New Brunswick to alleviate the financial and educational barriers faced by our participants. Studies such as this could allow researchers and service providers to evaluate the feasibility of implementing alternative models of ID delivery while centering the impacted populations in the process.

ID is not just a document; it is an entry point to full social participation. Addressing only one barrier, such as cost, without also tackling documentation requirements or address verification, will fail to meaningfully improve access for the most marginalized. As Wusinich et al. [28] and Gordon [3] argue, policy reform must consider the full spectrum of lived experience, including homelessness, past trauma, and systemic discrimination.

### 4.5. Limitations

The study had three main limitations. First, the recruitment methods may have introduced some selection bias. For the recruitment phase, we relied on our partnering organizations to support our recruitment efforts. However, the participants who had more time or were more involved with the organizations would have been more likely to participate in the study. Secondly, the data collected for this study were self-reported data that were not backed up by another original resource to confirm the information shared by the study participants. Lastly, while our recruitment efforts focused on having a diverse representation of our study population, we may not have fully captured it in this study sample.

## 5. Conclusions

This study highlights a critical policy tension: while service providers require ID to manage risk and eligibility, the rigidity of ID policies often makes the services themselves inaccessible to their intended beneficiaries. Our findings suggest that operational policies mandating ID need to be re-evaluated in light of their practical impact on equity and accessibility. Alternatives such as ID banks, biometric ID, trusted verifiers, or subsidized document support could meaningfully reduce barriers without sacrificing integrity.

In conclusion, our findings suggest that ID access is a foundational component of health and social inclusion yet remains inaccessible to many New Brunswickers due to a complex web of administrative, financial, and logistical barriers. These barriers disproportionately impact individuals with low socioeconomic status and precarious housing, preventing access to social determinants of health like financial stability, housing, and food supports. Community-informed alternatives such as ID banks and biometric systems offer promising avenues for reform but must be implemented with care, privacy protections, and equity at the core. Moving forward, governments and service providers must re-evaluate restrictive ID requirements and collaborate across sectors to pilot flexible, low-barrier solutions that center the voices of those most affected. Without such change, the cycle of exclusion persists alongside the preventable harms to individual and community well-being due to a lack of accessible ID.

## Figures and Tables

**Figure 1 ijerph-22-01552-f001:**
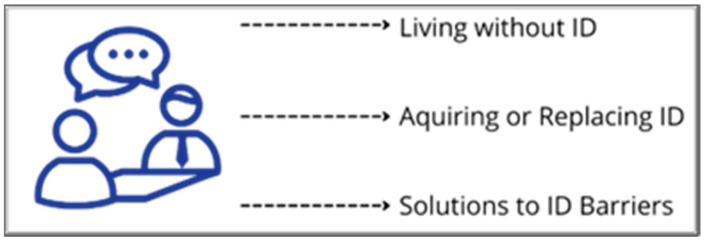
Three overarching themes emerged from the consumer interviews: living without an ID, acquiring or replacing an ID, and solutions to ID barriers.

## Data Availability

The datasets generated and/or analyzed for the current project are not available publicly due to privacy and confidentiality concerns but are available from the corresponding author on reasonable request.

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
