# Peer review of "Eyeing ID: Access to Identification as a Barrier to Banking and Other Social Determinants of Health"

_ijerph, 2025, doi:10.3390/ijerph22101552_

Round 1

Reviewer 1 Report

Comments and Suggestions for Authors

The study objective was to understand the implications of ID requirements, and the barriers to acquiring it through the lens of consumers. The authors used a mixed-methods, observational approach that included surveys and interviews. The methodology is appropriate. The writing quality is excellent. The findings contained surprises, and the information the research team discovered is useful.

My main substantive suggestion is that the authors more clearly state their preferred solution to the widespread lack of acceptable ID documents among low-income individuals, who need ID to access crucial public services. I also suggest that the authors discuss the generalizability of their findings. Do they apply only to people living in cities in New Brunswick?

Required changes:

  1. In the abstract, explain where the surveys and interviews were conducted. This is an international journal. Where the study was conducted could impact the generalizability of the findings. I learned that the respondents lived in New Brunswick, Canada, but readers should not have to read beyond the title and abstract to figure that out.
  2. What is a "scoping" review? l 57.
  3. Typo? : One participant also pointed out that “if you’re driving, the accept a photocopy of your driver’s license too.” l 362
  4. Please expand the explanation of ID Banks. I've never heard this term before.

Author Response

Thank you for taking the time to review and for your encouraging words and reflective feedback.

Reviewer comment 1: In the abstract, explain where the surveys and interviews were conducted. This is an international journal. Where the study was conducted could impact the generalizability of the findings. I learned that the respondents lived in New Brunswick, Canada, but readers should not have to read beyond the title and abstract to figure that out.

Response 1: Added a sentence in the abstract (manuscript has a comment indicating where) that introduces the location of the study. 

Reviewer comment 2: What is a "scoping" review? l 57.

Response 2: Removed "scoping" from "scoping review" to avoid confusion on definition. Note: The title of the work being referenced is "You Need ID to Get ID’: A Scoping Review of Personal Identification as a Barrier to and Facilitator of the Social Determinants of Health in North America" hence the initial use of the word "scoping".

Reviewer comment 3: Typo? : One participant also pointed out that “if you’re driving, the accept a photocopy of your driver’s license too.” l 362

Response 3: Addressed by changing "the" to "they".

Reviewer comment 4: Please expand the explanation of ID Banks. I've never heard this term before.

Response 4: Added a description of an ID bank by discussing the Kettle Society's operation in introduction. https://www.thekettle.ca/id-bank

Reviewer 2 Report

Comments and Suggestions for Authors

This is a valuable and timely manuscript addressing a neglected but highly relevant barrier to accessing the social determinants of health. It provides strong descriptive evidence and useful policy insights. However, the paper would improve by addressing the following:

  1. While the introduction reviews relevant literature, it could more explicitly situate the study within environmental health and public health Currently, it frames ID access primarily as a social justice issue.

Recommendation: Strengthen the connection to health outcomes (e.g., delays in healthcare access, stress, worsening chronic conditions) and explicitly discuss ID access as an environmental and structural determinant of health equity.

  1. The observational design is clear, but limitations are underemphasized. For example:
    1. More information can be provided about the selective recruitment methods.
    2. The urban-only focus limits generalizability to rural New Brunswick.
    3. Biases related to this study needs to be mentioned.

Recommendation: Expand the limitations section to explicitly acknowledge selection bias, recall bias (self-reported data), and limited representativeness. This would strengthen transparency.

  1. The survey analysis relies mainly on descriptive statistics. No inferential statistics are used to explore associations (e.g., between SES scores and difficulty obtaining ID).

Recommendation: Even with a modest sample, consider simple cross-tabulations. This would enrich the findings and demonstrate whether certain barriers disproportionately affect subgroups. Tables can be created for cross-tabulation which would really help readers to see the relationship between SES scores and difficulty obtaining ID.

  1. The quantitative and qualitative findings are presented in parallel but not fully integrated in the discussion.

Recommendation: Strengthen triangulation by directly linking numbers (e.g., “34.4% identified cost as a barrier”) with supporting narratives from interviews. This will enhance coherence.

  1. The manuscript references mainly North American studies but could benefit from broader international perspectives (e.g., national ID programs in Asian, and/or EU contexts).

Recommendation: Including global comparisons would strengthen the generalizability and show how other contexts have addressed similar barriers.

Author Response

Thank you for your thoughtful comments. We did our best to address below.

Reviewer 2 comment 1: While the introduction reviews relevant literature, it could more explicitly situate the study within environmental health and public health Currently, it frames ID access primarily as a social justice issue.

Recommendation: Strengthen the connection to health outcomes (e.g., delays in healthcare access, stress, worsening chronic conditions) and explicitly discuss ID access as an environmental and structural determinant of health equity.

Response 1: We added more details into introduction to address this feedback and additional reference.

Reviewer comment 2: The observational design is clear, but limitations are underemphasized. For example:

    1. More information can be provided about the selective recruitment methods.
    2. The urban-only focus limits generalizability to rural New Brunswick.
    3. Biases related to this study needs to be mentioned.

Recommendation: Expand the limitations section to explicitly acknowledge selection bias, recall bias (self-reported data), and limited representativeness. This would strengthen transparency.

Response 2: Expanded sentence in "Recruitment" section to explain the use of selective recruitment methods, and added a limitations subsection at the end of discussion section. 

Reviewer comment 3: The survey analysis relies mainly on descriptive statistics. No inferential statistics are used to explore associations (e.g., between SES scores and difficulty obtaining ID).

Recommendation: Even with a modest sample, consider simple cross-tabulations. This would enrich the findings and demonstrate whether certain barriers disproportionately affect subgroups. Tables can be created for cross-tabulation which would really help readers to see the relationship between SES scores and difficulty obtaining ID.

Response 3: Thank you for the suggestion to cross-tabulate the data to explore associations between SES scores and how difficult is was to obtain ID. We agree that this an important analysis to add to our findings. However due to the time constrains we are not able to reanalyze the data to include this recommendation but will consider in a future revision in necessary.

Reviewer comment 4: The quantitative and qualitative findings are presented in parallel but not fully integrated in the discussion.

Recommendation: Strengthen triangulation by directly linking numbers (e.g., “34.4% identified cost as a barrier”) with supporting narratives from interviews. This will enhance coherence.

Response 4: Three separate narrative quotes were added which have been labelled on the manuscript (line 417, 451, 482) .

Reviewer comment 5: The manuscript references mainly North American studies but could benefit from broader international perspectives (e.g., national ID programs in Asian, and/or EU contexts).

Recommendation: Including global comparisons would strengthen the generalizability and show how other contexts have addressed similar barriers.

Response 5: Added comment and world bank reference (line 504 )

Reviewer 3 Report

Comments and Suggestions for Authors

Peer Review for IJERPH

Thank you for submitting this research. The introduction reads well, and the ID literature is adequate. I have some recommendations that may improve the manuscript further:

  1. The theory on social determinants of health (SDOH) can be improved significantly; currently only Sanders is cited. What about other work on SDOH? There is numerous.

E.g. https://books.google.com/books?hl=en&lr=&id=CQkiDAAAQBAJ&oi=fnd&pg=PP1&dq=social+determinants+of+health+in+canada&ots=QmVczeVl5o&sig=DMjeKjTMHiPymoM3TZUKeEG69RI

  1. How was data from survey monkey extracted? What file format was it and how did the team reconcile this into SPSS e.g. did data need to be cleaned up?
  2. How was coding of interview data conducted – manually or was software used? If yes, which software?
  3. Why was only New Brunswick sampled? Why not all of Canada?
  4. When paper copies of the survey were administered, were they mailed back to the research team by the participants or did the partner organizations scan/email/fax/mail them?
  5. The term “government issued Photo ID” needs to be defined. Is this provincial government ID? Federal government ID e.g. passport? Birth certificate? Immigration document? In lines 67 to 75, provincial photo ID also needs to be defined, as there are instances of mentioning and conflating provincial with federal ID but provincial ID would include driver’s license…
  6. What was the exclusion/inclusion criteria?
  7. What did you mean “financial inclusion” in line 103? This term needs to be defined…Participants were recruited based on “barriers […] to financial inclusion” does not make sense.
  8. Please attach a copy of the questionnaire – was race/migrant status/birth status collected or no?

Thank you for the opportunity to review this work.

Author Response

Thank you for taking the time to review our work and providing thoughtful and constructive ideas on how to improve the manuscript. We did our best to respond below:

Reviewer comment 1: The theory on social determinants of health (SDOH) can be improved significantly; currently only Sanders is cited. What about other work on SDOH? There is numerous. E.g. https://books.google.com/books?hl=en&lr=&id=CQkiDAAAQBAJ&oi=fnd&pg=PP1&dq=social+determinants+of+health+in+canada&ots=QmVczeVl5o&sig=DMjeKjTMHiPymoM3TZUKeEG69RI

Response 1: Thank you your commitment to being clear this effects the social determinants of health. The introduction paragraph beginning with "In Canada, ID is frequently required to..." is a full paragraph with 6 references which addresses this. We could not access the entire book as a reference to be appropriate in the time period but did try to be more explicit that these were the determinants of health.

Reviewer comment 2: How was data from survey monkey extracted? What file format was it and how did the team reconcile this into SPSS e.g. did data need to be cleaned up?

Response 2: This information was added to the manuscript (line 157)

Reviewer comment 3: How was coding of interview data conducted – manually or was software used? If yes, which software?

Response 3: Added "Manually" to Data analysis section "Initial coding was manually conducted by several members of the research team independently following Braun et al.’s 2018 thematic analysis framework [19]." (line 160)

Reviewer comment 4: Why was only New Brunswick sampled? Why not all of Canada?

Response 4: Our ethics approval from Horizon Health Network (HHN), was only for the New Brunswick data. Also the pandemic played a role where we could not travel outside the province and had to do a lot of our recruitment/survey collection in person given that we went to local community groups to seek participants and do surveys on paper.

Reviewer comment 5: When paper copies of the survey were administered, were they mailed back to the research team by the participants or did the partner organizations scan/email/fax/mail them?

Response 5: The entered the responses into survey monkey on behalf of the respondent. 

Reviewer comment 6: The term “government issued Photo ID” needs to be defined. Is this provincial government ID? Federal government ID e.g. passport? Birth certificate? Immigration document? In lines 67 to 75, provincial photo ID also needs to be defined, as there are instances of mentioning and conflating provincial with federal ID but provincial ID would include driver’s license…

Response 6: This was completed.

Reviewer comment 7: What was the exclusion/inclusion criteria?

Response 7: "All study participants voluntarily consented and were required to be 19 years of age or older, living in the province of New Brunswick" (line 117 )We did aim our study recruitment at community programs that we knew supported the participants we aimed to reach (aka lower SES, living without ID, etc) but those weren't necessarily criteria. Whoever participated was included as long as they were living in NB and over 19. 

Reviewer comment 8: What did you mean “financial inclusion” in line 103? This term needs to be defined…Participants were recruited based on “barriers […] to financial inclusion” does not make sense.

Response 8: We have added this definition to the introduction (line 103 ).

Reviewer comment 9: Please attach a copy of the questionnaire – was race/migrant status/birth status collected or no?

Response 9: We have added an option to contact the author for a copy of the questionnaire - it is 12 pages long and thus out of the word count allotment for the manuscript. To answer the second part of your question, we asked: 

Are you a Canadian citizen or permanent resident? 

[Yes/No

Do you identify as Indigenous? 

[Yes/No

Round 2

Reviewer 2 Report

Comments and Suggestions for Authors

Comments have been addressed.

Author Response

Thank you for accepting our revisions and for the time it took to review. 

Reviewer 3 Report

Comments and Suggestions for Authors

Thank you for submitting this research. The introduction reads well, and the ID literature is adequate. I have some recommendations that may improve the manuscript further:

The theory on social determinants of health (SDOH) can be improved significantly; currently only Sanders and LeBron are cited.

The authors have addressed most of the concerns except citing more on social determinants of health literature e.g. what about income and income distribution instead of "financial services" -  https://link.springer.com/content/pdf/10.1186/1475-9276-11-41.pdf ? 

e.g. housing -  https://www.researchgate.net/profile/Toba-Bryant/publication/236144262_The_current_state_of_housing_in_Canada_as_a_social_determinant_of_health/links/575ea93208aed884621b601c/The-current-state-of-housing-in-Canada-as-a-social-determinant-of-health.pdf

e.g. transportation - https://www.sciencedirect.com/science/article/pii/S0890406522000056 

e.g. food  - http://irpp.org/wp-content/uploads/sites/2/assets/po/bank-mergers/mcintyre.pdf 

Thank you for the opportunity to review this work.

Author Response

Thank you for this feedback and the encouragement to further strengthen the section on the SDOH - We have added an extension to the paragraph of the introduction with three more references provided particularly going into more detail about financial and housing insecurity and transportation challenges. Lines 101-118.